# Synthetic Multi-inversion Time Magnetic Resonance Images from Routine Clinical Contrasts

**Savannah P. Hays**[1] (ID)                         SHAYS6@JHU.EDU
[1] *Image Analysis and Communications Laboratory, Department of Electrical and Computer Engineering, Johns Hopkins University, USA*

**Lianrui Zuo**[2] (ID)                        LIANRUI.ZUO@VANDERBILT.EDU
[2] *Department of Electrical and Computer Engineering, Vanderbilt University, USA*

**Anqi Feng**[1] (ID)                          AFENG11@JHU.EDU
**Yihao Liu**[2] (ID)                         YIHAO.LIU@VANDERBILT.EDU
**Blake E. Dewey**[3] (ID)                         BLAKE.DEWEY@JHU.EDU
[3] *Department of Neurology, Johns Hopkins University School of Medicine, USA*

**Jiachen Zhuo**[4] (ID)                      JZHUO@SOM.UMARYLAND.EDU
[4] *Department of Diagnostic Radiology and Nuclear Medicine, University of Maryland School of Medicine, USA*

**Ellen M. Mowry**[3] (ID)                          EMOWRY1@JHMI.EDU
**Scott D. Newsome**[3] (ID)                        SNEWSOM2@JHMI.EDU
**Jerry L. Prince**[1] (ID)                          PRINCE@JHU.EDU
**Aaron Carass**[1] (ID)                        AARON_CARASS@JHU.EDU

## Abstract

Visualization of subcortical gray matter is essential in neuroscience and clinical practice, particularly for disease understanding and surgical planning. Multi-inversion time (multi-TI) $T_1$-weighted ($T_1$-w) magnetic resonance (MR) imaging improves visualization of these structures, but is not available in common public datasets. We present SyMTIC (Synthetic Multi-TI Contrasts), a physics-informed deep learning framework that generates multi-TI images from routinely acquired $T_1$-w, $T_2$-weighted ($T_2$-w), and FLAIR images. SyMTIC estimates quantitative longitudinal relaxation time ($T1$) and proton density ($\rho$) maps, which are used within an inversion recovery signal model to synthesize MR images at arbitrary inversion times. This formulation enables flexible contrast generation beyond fixed targets such as FGATIR. On an in-domain dataset ($N = 23$), SyMTIC produces high-quality synthetic images, achieving a PSNR/SSIM of $45.70 \pm 5.67$ / $0.9970 \pm 0.0029$ for MPRAGE and $27.60 \pm 2.27$ / $0.7906 \pm 0.0535$ for FGATIR. The synthesized contrasts improve visualization of subcortical structures and support downstream tasks such as thalamic segmentation. Additionally, by incorporating HACA3-based harmonization and imputation, SyMTIC generalizes to out-of-domain datasets and scenarios with missing input contrasts. These results demonstrate that SyMTIC provides a practical and flexible solution for enhancing MR image contrast and analysis using standard clinical acquisitions.

**Keywords:** MRI, image synthesis, brain

## 1. Introduction

Visualization of subcortical structures is critical for neurological disease analysis and surgical planning (Power et al., 2015), particularly in applications such as deep brain stimulation (DBS) (Krauss et al., 2020). Specialized magnetic resonance (MR) image contrasts such as fast gray matter acquisition T1 inversion recovery (FGATIR) improve visualization of thalamic nuclei but are not commonly available in public datasets such as OASIS (LaMontagne et al., 2019) and ADNI (Mueller et al., 2005). The 3D magnetization prepared rapid acquisition with gradient echo (MPRAGE) image (Mugler III and Brookeman, 1990) is a high resolution, fast acquisition that is more commonly acquired. FGATIR images provide improved contrast for visualization and segmentation of subcortical structures compared to MPRAGE (Sudhyadhom et al., 2009). Recent work has explored synthesizing FGATIR images using deep learning (Umapathy et al., 2021; Moya-Sáez et al., 2021; Tohidi et al., 2023; Jog et al., 2017; Hays et al., 2024) but existing approaches are typically limited to fixed contrasts or require segmentation. Some work relies on specialized acquisitions such as the MP2RAGE sequence (Middlebrooks et al., 2018). To address these limitations, we propose SyMTIC (Synthetic Multi-TI Contrasts), a deep learning framework for generating multi-inversion time (multi-TI) images from routinely acquired MR contrasts. Our physics-informed synthesis method first produces $T1$ and proton density ($\rho$) maps from $T_1$-weighted ($T_1$-w), $T_2$-weighted ($T_2$-w), and fluid-attenuated inversion recovery (FLAIR) images. These parameter maps are then used for the generation of arbitrary TI contrasts, not limited to FGATIR. This paper presents a focused summary of our recent journal work on SyMTIC (Hays et al., 2026b), highlighting the core methodology and in-domain evaluation, while additional experiments and broader validation are detailed in the full study. Our model is open source and publicly available from https://github.com/shays15/symtic.git.

## 2. Methods

SyMTIC combines deep learning with MR physics to enable flexible contrast synthesis as shown in Figure 1. The model takes commonly acquired MR images (T1w, T2w, FLAIR) as input and predicts quantitative $T1$ and $\rho$ maps. These parameter maps are then used in an inversion recovery signal model to generate MR images at arbitrary TIs. This formulation enables continuous control over image contrast rather than restricting synthesis to a single target image. To improve robustness in real-world settings, we incorporate harmonization and contrast imputation using HACA3 (Zuo et al., 2023) or HACA3$^+$ (Hays et al., 2026a), allowing the model to operate with either out-of-domain data or missing input modalities.

## 3. Experiments and Results

We evaluated SyMTIC on an in-domain test set ($N = 23$). An illustrative example is shown in Fig. 2, comparing synthetic outputs with the corresponding ground truth images (acquired images and computed parameter maps). Quantitative evaluation was performed using peak signal-to-noise ratio (PSNR) and structural similarity index (SSIM), computed over foreground voxels only. For the synthetic MPRAGE images, SyMTIC achieved a PSNR of $45.70 \pm 5.67$ and SSIM of $0.9970 \pm 0.0029$. For FGATIR, the model achieved $27.60 \pm 2.27$ PSNR and $0.7906 \pm 0.0535$ SSIM. For the estimated parameter maps, PSNR/SSIM were

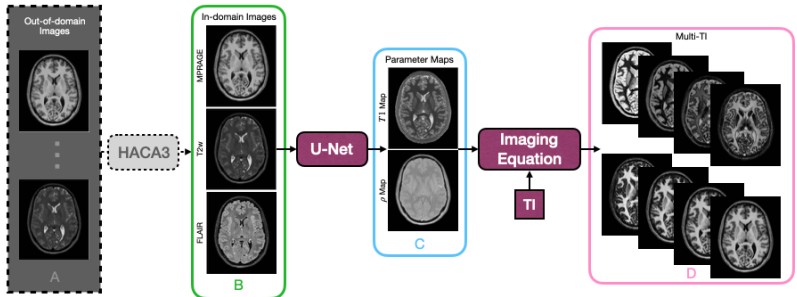

Figure 1: Overview of SyMTIC. **(A)** HACA3 is used for harmonization and/or imputation of out-of-domain images. **(B)** Required in-domain images input to the U-Net model. **(C)** Synthesis of the $T1$ and $\rho$ parameter maps. **(D)** Calculation of multi-TI images using the imaging equation with specific TIs.

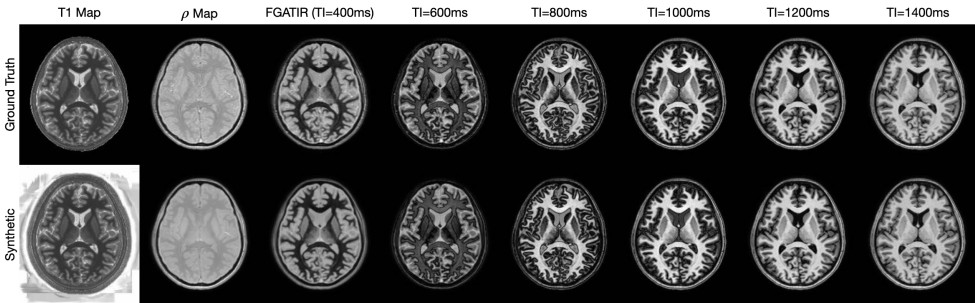

Figure 2: Ground truth images (top row) for a test subject with the synthetic images (bottom row) using the SyMTIC model.

$23.75 \pm 1.73$ / $0.8088 \pm 0.0319$ for $T1$ and $29.82 \pm 2.14$ / $0.8516 \pm 0.0365$ for $\rho$. These results indicate that SyMTIC accurately reconstructs both image contrast and underlying quantitative parameters, with particularly high fidelity in the synthesized MPRAGE images and consistent performance across subjects.

## 4. Discussion and Conclusion

This paper summarizes our recent journal work on SyMTIC, which introduces a flexible framework for MR image synthesis based on quantitative parameter estimation. Unlike prior approaches that generate a single contrast, SyMTIC enables continuous contrast synthesis across TIs, providing greater flexibility for clinical and research applications. A key strength of the approach is its ability to operate on standard clinical inputs while maintaining robustness to domain shifts and missing modalities by using HACA3 or HACA3[+]; which makes it particularly suitable for multi-site datasets. By enabling arbitrary contrast generation, SyMTIC provides a practical tool for enhancing visualization of subcortical structures in clinical neuroimaging.

## Acknowledgments

This research is partially supported by the Johns Hopkins University Percy Pierre Fellowship (Hays) and the National Science Foundation Graduate Research Fellowship under Grant No. DGE-2139757 (Hays). Development is partially supported by FG-2008-36966 (Dewey), CDMRP W81XWH2010912 (Prince), NIH R01EB036013 (Prince), NIH R01 CA253923 (Landman), NIH R01 CA275015 (Landman), the National MS Society grant RG-1507-05243 (Pham) and Patient-Centered Outcomes Research Institute (PCORI) grant MS-1610-37115 (Newsome and Mowry). The statements in this publication are solely the responsibility of the authors and do not necessarily represent the views of the Patient-Centered Outcomes Research Institute (PCORI), its Board of Governors or Methodology Committee.

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
