# OpenReview forum: "Synthetic Multi-inversion Time Magnetic Resonance Images from Routine Clinical Contrasts"
_MIDL.io/2026/Short_Papers — MIDL 2026 - Short Papers Poster_

### Official Review · Reviewer_R38R · 2026-05-03

**Rating:** 5
**Confidence:** 5

**Review:**

- The research question is important, the approach is novel, and the performance is seemingly impressive, although baseline comparisons would improve the presentation of the results.
- Another MIDL publication has already tackled a similar problem, transferring a single MR signal into another signal using their respective signal equations, I would recommend the authors this publication: [https://2021.midl.io/proceedings/simko21.pdf]

**Summary:**

The paper presents SyMTIC, a deep learning approach that decomposes a combination of T1w, T2w and FLAIR input images into a T1 and density maps, which can be used to reconstruct any arbitrary MR sequence. The model is later evaluated on two datasets through PSNR and SSIM as well as qualitatively. Open source code is also available together with the model weights.

**Strengths:**

- The research question is important and the proposed appproach is novel and impressive.
- The paper is well written, and the methodology is easy to follow.
- The open source code is much appreciated.
- Although more information would certainly benefit the presentation of the results, within the scope of a short paper the authors have managed to focus on the essentials.

**Weaknesses:**

- It is hard to assess the quality of SyMTIC without any baseline. In the current submission, the claim that the "results indicate that SyMTIC accurately reconstructs [...]" is a bit misleading. Without a baseline, it would be more constructive to show qualitative results, limitations, etc. Or even better, introducing a baseline for comparison.
- It is a bit unclear exactly what datasets have been used for the project, and whether they are publicly available, or not, and whether they have references/links or not.

**Justification Of Rating:**

The paper presents a deep learning framework for reconstructing any custom MR image synthesis based on quantitative parameter estimation. The qualitative evaluation looks impressive, and although the 3 page limit is a clear limitation of presenting a work of this scale, I believe it suits the conference well.

---

### Decision · Program_Chairs · 2026-05-08

Accept (Poster)